# Follicle-Stimulating Hormone (FSH) Action on Spermatogenesis: A Focus on Physiological and Therapeutic Roles

**DOI:** 10.3390/jcm9041014

**Published:** 2020-04-03

**Authors:** Daniele Santi, Pascale Crépieux, Eric Reiter, Giorgia Spaggiari, Giulia Brigante, Livio Casarini, Vincenzo Rochira, Manuela Simoni

**Affiliations:** 1Department of Biomedical, Metabolic and Neural Sciences, University of Modena and Reggio Emilia, 41126 Modena, Italy; santi.daniele@gmail.com (D.S.); giulia.brigante@unimore.it (G.B.); livio.casarini@unimore.it (L.C.); vincenzo.rochira@unimore.it (V.R.); 2Unit of Endocrinology, Department of Medical Specialties, Azienda Ospedaliero-Universitaria of Modena, 41126 Modena, Italy; giorgia.spaggiari87@gmail.com; 3Physiologie de la Reproduction et des Comportements (PRC), Institut National de Recherche pour l’Agriculture, l’Alimentation et l’Environnement (INRAE), Centre National de la Recherche Scientifique (CNRS), Institut Français du Cheval et de l’Equitation (IFCE), Université de Tours, 37380 Nouzilly, France; pascale.crepieux@inrae.fr (P.C.); eric.reiter@inra.fr (E.R.)

**Keywords:** FSH, spermatogenesis, male infertility

## Abstract

Background: Human reproduction is regulated by the combined action of the follicle-stimulating hormone (FSH) and the luteinizing hormone (LH) on the gonads. Although FSH is largely used in female reproduction, in particular in women attending assisted reproductive techniques to stimulate multi-follicular growth, its efficacy in men with idiopathic infertility is not clearly demonstrated. Indeed, whether FSH administration improves fertility in patients with hypogonadotropic hypogonadism, the therapeutic benefit in men presenting alterations in sperm production despite normal FSH serum levels is still unclear. In the present review, we evaluate the potential pharmacological benefits of FSH administration in clinical practice. Methods: This is a narrative review, describing the FSH physiological role in spermatogenesis and its potential therapeutic action in men. Results: The FSH role on male fertility is reviewed starting from the physiological control of spermatogenesis, throughout its mechanism of action in Sertoli cells, the genetic regulation of its action on spermatogenesis, until the therapeutic options available to improve sperm production. Conclusion: FSH administration in infertile men has potential benefits, although its action should be considered by evaluating its synergic action with testosterone, and well-controlled, powerful trials are required. Prospective studies and new compounds could be developed in the near future.

## 1. Introduction

The pituitary gland regulates human reproduction through the refined, combined action of the follicle-stimulating hormone (FSH) and the luteinizing hormone (LH) on the gonads. Upon transport in the bloodstream, FSH reaches the gonads, stimulating follicle development in females and spermatogenesis in males.

While the FSH role on reproductive physiology in both sexes is clear, its therapeutic use for the treatment of infertility remains doubtful, especially in men. FSH is mostly used in the therapeutic context of assisted reproduction techniques (ART) for inducing multi-follicular growth in women, while the efficacy of FSH administration in infertile men is unclear. In clinical pictures characterized by deficient levels of FSH, such as male hypogonadotropic hypogonadism (HH), restoration of FSH action improves quantitative and qualitative sperm parameters and, therefore, fertility [1]. The therapeutic approach to male HH aims at restoring the physiological functioning of the testis and can be obtained by the administration of gonadotropin-releasing hormone (GnRH), which mediates the endogenous, pituitary production of both FSH and LH. Proper serum gonadotropin levels may also be achieved by exogenous administration of these hormones, typically human chorionic gonadotropin (hCG), which, acting on the same membrane receptor for LH, the LHCGR, was assumed to be identical to LH for long time [2]. hCG administration may be supported by adding FSH activity with either urinary-derived or recombinant follitropins [3].

In HH patients, the increase of serum gonadotropin levels is suggested to be beneficial for male fertility, since the improvement of sperm parameters would have a positive impact on pregnancy rate. Although the full restoration of normal semen parameters is generally not achieved in HH men (depending on HH causes) [4] and not even necessary to obtain a pregnancy, gonadotropin treatment is sufficient to stimulate spermatogenesis [1] and provides a rationale for using, even if empirically, FSH, with or without hCG, in male idiopathic infertility, which is considered a form of “functional” hypogonadism. By analogy with the therapeutic benefits observed in HH males, exogenous FSH administration has been proposed in men with altered sperm production and FSH levels within the normal range, assuming its capacity to increase the spermatogenic output [5].

In the present review, we evaluate the physiological role of FSH in spermatogenesis, highlighting its synergic action with testosterone. These concepts are fundamental to understand the potential pharmacological benefits of FSH administration in clinical practice.

## 2. Physiological Control of Spermatogenesis

Spermatogenesis occurs within testicular seminiferous tubules in a stepwise fashion, requiring autocrine, paracrine, and endocrine stimuli that are controlled by both FSH and LH actions [6,7]. FSH is a glycoprotein formed by two subunits: alpha, shared with other glycoprotein hormones, and beta (FSHβ). The human FSHβ is encoded by the *FSHB* gene located on chromosome 11p21 [8]. Through interaction with its receptor (FSHR) [9], FSH acts on its unique target in male cells, namely, the Sertoli cells, located at the basis of the seminiferous tubules of the testis [10,11]. These cells create a niche in which spermatogonia proliferate and mature [10,11]. Sertoli cells are connected together and to neighboring germ cells by gap junctions, permitting metabolite exchanges. Importantly, tight junctions, located at their basis, form the blood–testis barrier, isolating meiotic and post-meiotic germ cells from the bloodstream. In Leydig cells, LH stimulates testosterone production through the interaction with its specific receptor, the LHCGR [12]. Testosterone achieves 50–100-fold higher concentrations within the testis than in peripheral circulation [12]. All these aspects point out the importance of the testis environment for the support and maintenance of the spermatogenetic function [13,14].

The physiological role of FSH in spermatogenesis regulation was evaluated in different animal models. In rodents, FSH determines the final Sertoli cell number at puberty, by stimulating cell proliferation during fetal and neonatal life, whereas, in primates, this mitotic function is observed during neonatal and peri-pubertal stages [15]. Consistently, early in life, FSH stimulates the transcription of genes involved in both DNA replication and cell cycle regulation [16].

In humans, the *FSHR* is first expressed during the second half of gestation, but its activation occurs after the onset of FSH secretion in the newborn [17]. Then, the peri-pubertal rise of FSH stimulates Sertoli cell proliferation [15]. In adulthood, FSH drives Sertoli cells to produce regulatory molecules and nutrients required for spermatogenesis [18]. In particular, FSH activates the transcription of genes involved in metabolic homeostasis and supports germ cell functions [16], with the synthesis of retinoic acid, lactate, type 2 plasminogen activator, as well as fatty acid metabolism and mitochondrial biogenesis [19,20]. FSH circulating levels correlate directly with Sertoli cell number and testicular volume in adults [21].

Beyond the known FSH action on Sertoli cell proliferation, the precise role of this gonadotropin in spermatogenesis remains largely unclear. Genetically modified mouse models have been useful to better understand how FSH regulates spermatogenesis. In particular, in the adult mouse testis, FSH stimulates Sertoli cells to produce anti-apoptotic survival factors and adhesion molecules, facilitating germ cell maturation [22,23]. However, the absolute lack of FSH or *FSHR*, despite a reduction in Sertoli cell number, does not lead to azoospermia nor sterility [24,25]. In fact, *Fshb* or *Fshr* knockout (KO)-mice have reduced, yet persisting, sperm production. More precisely, the lack of FSH action leads to reduced Sertoli cell numbers but germ cell maturation persists, despite a decreased number of spermatogonia and spermatocytes. With this in mind, animal models suggest that FSH is required to elicit Sertoli cell proliferation and to maintain germ cell numbers, probably through the ability of Sertoli cells to nurture germ cells, whereas its action is dispensable to complete spermatogenesis. Since the endocrine regulation of gonadal functions in mice could differ from humans, data from rodent models should be considered in their context. In some cases, inactivating homozygous *FSHR* mutations in humans were associated with infertile male phenotypes [26], although these mutations are very rare, and the issue is a matter of debate [25,27] and merits additional investigations. However, it is also worth noting that the A189V inactivating mutation is related to male oligozoospermia/subfertility, not necessarily azoospermia [28].

Wide evidence is available in the scientific literature, demonstrating the central role of LH in supporting spermatogenesis via induction of intratesticular testosterone production. Indeed, *LH receptor (Lhr)*-KO mice models have no testosterone production and exhibit an impaired progression of spermatogenesis [29] that is restored after testosterone replacement [30]. Thus, unlike FSH, the absence of LH-dependent testosterone production leads to azoospermia, suggesting that testosterone is strictly required for sperm production. These experimental models suggest that a quantitatively and qualitatively normal spermatogenesis requires the action of intratesticular testosterone in synergy with FSH.

The existence of synergism between testosterone and FSH is also suggested by in vitro studies, since both testosterone and FSH regulate the expression of genes involved in blood–testis barrier development and functions, while FSH supports the organization of tight junctions and ectoplasmic specializations [16,31,32], suggesting that the synergistic action of FSH and testosterone regulate different levels of the spermatogenic processes [33]. New insights on the nature of this synergy came from the *Fshr*-D580H mutation in *Lhr*-KO background mutant mice [34]. The *Fshr*-D580H mutation, resulting in constitutive *Fshr* activation, reversed the azoospermia due to missing LH action obtained by the deletion of the *Lhr* gene, combined with the blockade of the residual testosterone activity by the antiandrogen flutamide [34]. These data demonstrate that the constitutive activation of *Fshr*-mediated signaling may overcome the absence of both LH- and testosterone-mediated pathways. Actually, the continuous hyperactivation of FSH-like signals sustains the transcription of androgen-dependent genes and activates the spermatogenic pathway in a LH- and testosterone-independent way [34].

Overall, the existing literature in other animal models suggests that signals induced by gonadotropins may be non-specific and redundant since in some species spermatogenesis occurs in the absence of FSH [24,25], indicating that the action of intratesticular testosterone is sufficient for supporting sperm production in physiological conditions. Although these effects might occur only in specific experimental settings [35,36], FSH is supposed to play pleiotropic roles on gametogenesis. This FSH-dependent gonadal regulation would be sex-specific, since the hormone is instead indispensable for follicular growth beyond the pre-antral stage in females [25].

In humans, the physiological framework in which FSH acts is even more complex [37], and the effective action of the hormone on spermatogenesis remains poorly understood. The few data describing the effects of impaired FSH action in the human are provided by *FSHR* [28] and *FSHB* inactivating mutations [38]. The rare *FSHR* mutations described in men suggest they lead to subfertility with testosterone levels within the physiological range and reduced spermatogenesis, but not necessarily to azoospermia. This result reflects the clinical picture in *Fshr*-KO mice models, suggesting that spermatogenesis may occur in the absence of proper FSH action also in humans, while azoospermia occurs in all cases of inactivating *FSHB* mutations described so far [38]. However, considering the paucity of *FSHB* and *FSHR* mutations described in men, we are currently unable to clarify the exact role of FSH for human spermatogenesis.

Human inactivating *LHCGR* or *LH beta (LHB)* mutations are linked to 46, XY disorders of sexual development (DSD) phenotypically ranging from the complete feminization to less severe or no sexual abnormalities [39]. In certain cases, LH signaling defects not fully inhibiting the hormone action lead to oligozoospermia and low intratesticular testosterone levels, but not azoospermia [40,41]. Thus, in the presence of physiological FSH production, very low levels of LH could be sufficient for sustaining intratesticular testosterone and sperm production in humans. These data may be suggestive of a possible FSH and LH synergistic action based on the redundancy of the gonadotropins’ signaling machinery [42,43]. On the other hand, human activating *LHCGR* mutations lead to increased testosterone levels associated with continuous production of spermatozoa, despite reduced FSH levels [42]. It may be speculated that, during evolution, spermatogenesis has been maintained under partial dependence of both FSH and LH, at least in certain species [44], ensuring sperm production despite the loss of the action of one gonadotropin.

Information about the physiological role of FSH in spermatogenesis comes from mammalian models of hemicastration, including primates, where a significantly increased volume of the contralateral testis occurs [45,46,47,48,49]. Considering that the adult Sertoli cell numbers do not change, the increase in testicular size is reasonably due to the growing amount of germ cells [49]. Unilateral orchiectomy results in inhibin B decrease, resulting in endogenous FSH rise and support of sperm production through increased stimulation of B spermatogonia proliferation [49]. Similar data were described in men after removal of a testis with malignant cells, where contralateral testicular hypertrophy occurred [45]. In this setting, testicular hormonal secretion and sperm production returned to physiological levels within 120 days after hemicastration, confirming the compensatory increase of spermatogenesis in the remaining testis [45]. These data are corroborated by studies of patients affected by pituitary FSH-secreting adenomas, which showed elevated testicular enlargement likely due to FSH over-stimulation, even when occurring after puberty [50].

In summary, human spermatogenesis (Figure 1) is physiologically regulated both by FSH action and, synergistically, by LH-dependent intra-testicular testosterone, which, in turn, acts in Sertoli cells through the androgen nuclear receptor (AR). These mechanisms are exerted through the activation of partially overlapping intracellular signaling pathways since the absence of either gonadotropin does not necessarily determine azoospermia. However, qualitatively and quantitatively adequate spermatogenesis requires FSH action and high intratesticular testosterone levels, acting in additive and synergistic ways [51]. These data provide new perspectives in the therapeutic approaches to male infertility, similar to what is currently done in women undergoing ART [52].

## 3. Mechanism of Action of FSH in the Sertoli Cell

Many studies have aimed at deciphering the signaling pathways induced by FSH from birth to puberty in order to gain insights into the dynamics that govern the switch between the hormone mitotic and differentiating functions. Coupling of the *FSHR* to the G protein α subunit (Gα_s_)/cyclic adenosine monophosphate (cAMP)/cAMP-dependent protein kinase A (PKA)/cAMP-responsive elements binding protein (CREB) signaling pathway has been acknowledged as the sole effector mechanism of FSH for more than forty years [53]. In Sertoli cells, FSH regulates the intracellular concentration of cAMP, resulting from the equilibrium between its synthesis by adenylate cyclase and degradation rate by phosphodiesterases (PDE) [54,55,56]. The intracellular concentration of second messengers is also fine-tuned by desensitization of the *FSHR*, which occurs within minutes upon FSH exposure [57,58], and it results in uncoupling of the ligand-bound receptor from G proteins. In vitro treatment of Sertoli cells by FSH leads to the recruitment at the receptor of kinases that target G protein-coupled receptor (GPCR), namely the G protein-coupled receptor kinases (GRKs) [57,58]. β-arrestins are then recruited to the *FSHR* as the result of both phosphorylation and agonist-induced conformational changes of the receptor. β-arrestins promote internalization by virtue of their ability to interact with components of the clathrin-coated pits. This desensitization process is of clear physiological relevance because, in the living organism, Sertoli cells are permanently exposed to FSH, being the receptor localized at the basal pole of the cell, in contact with the basal lamina and blood capillaries (Figure 2).

In the early 2000s, a number of discoveries have highlighted that FSH actually induces a complex network of molecular interactions within the cell, which largely exceeds G protein-dependent signaling. The coupling of the *FSHR* to the Gα_s_/cAMP/PKA signaling pathway is now viewed as one among several mechanisms contributing to the activation of the whole hormone-induced signaling network, where cAMP-dependent and cAMP-independent signaling co-exist. In Sertoli cells, promiscuity of G protein coupling has been demonstrated so far as the only way to promote cAMP-independent signaling. For example, the *FSHR* couples not only to Gα_s_ but also to Gα_i_, which is involved in the post-natal mitotic response of Sertoli cells to FSH via extracellular-regulated kinase (ERK)/mitogen-activated protein kinase (MAPK)-dependent signaling [59]. These data are supported by recent evidence that FSH limits the expression of Gα_s_, which fails to downregulate Gα_i_ activity in infant primates [60]. In this line, a dynamic computational model predicted that PKA inhibition does not compromise phosphatidylinositide(3,4,5) trisphosphate (PIP3) production in pre-pubertal rats [61]. PKA-independency of this pathway has been validated experimentally by studying the activation of protein kinase B (PKB/Akt), a target of PIP3 kinase (PI3K) in Sertoli cells [62]. ERK and Akt are regulated by FSH in an opposite manner during Sertoli cell post-natal development [59,61]. The tyrosine kinase receptor phosphatase Src homology 2-containing phosphotyrosine phosphatase (Shp2) is likely involved in this dual regulation of ERK and Akt by FSH, and disruption of this pathway in Shp2 knock-out mice leads to infertility because of abnormalities in the blood–testis barrier and premature exhaustion of the germ cell population [63]. Numerous studies employing immortalized cell lines have opened the possibility that β-arrestins could be a major scaffold for many signaling modules of GPCRs in general [64], and of the *FSHR* in particular [65,66,67]. Their action is important because they sequentially dictate the kinetics of ERK activation profile, which is more sustained than the transient Gs-mediated ERK activation [65]. In contrast, β-arrestins and Gα_s_ act cooperatively to stimulate the ribosomal protein S6 kinase beta-1 (p70S6K) in response to FSH in a complex including one of the enzyme substrates (ribosomal protein S6) [68]. Although p70S6K and β-arrestin 1 are identified in common protein complexes in Sertoli cells, to date, the role that β-arrestins could play as regulators of the kinetics and spatial organization of FSH-dependent signalling network has not been conclusively proven in these cells. However, the fact that oligozoospermic men bearing the inactivating Ala189Val mutation in the *FSHR* retain β-arrestin-dependent but cAMP-independent signaling [69] suggests that β-arrestins may play a role in supporting fertility.

Strikingly, in Sertoli cells, second messenger levels appear to be developmentally regulated, with the efficacy of FSH-induced cAMP production rising from birth to puberty [59], whereas the sensitivity to FSH of PIP3 production decreases over time [70]. For example, in FSH-stimulated Sertoli cells, p70S6K is regulated by a subtle interplay between PKA- and PI3K-dependent signaling that operates in a developmentally-regulated manner [70]. PIP3-dependent molecular events, such as the mammalian target of rapamycin (mTOR) and proline-rich Akt substrate of 40 kDa (PRAS-40) phosphorylation, appear to be involved in the mitogenic response to FSH, whereas 5′ AMP-activated protein kinase (AMPK) and phosphatase and tensin homolog (PTEN) counteract this effect [71,72]. However, the mechanisms whereby the *FSHR* couples to PIP3 regulation is still unclear.

The FSH signaling network dictates the protein content of gonadal cells through long-term transcriptional regulations directly affected by transcriptional factors or chromatin remodeling, and through short-term post-transcriptional regulations at the level of RNA messenger (mRNA) translation and micro RNA (miRNA) turn-over. DNA microarray analyses of FSH-treated prepubertal rat Sertoli cells in vitro revealed effects on the steady-state level of many target mRNAs, as early as after 2 h of treatment [73]. Suppressing FSH by a four-day passive immunization with neutralizing Abs led to the identification of hormone-regulated transcripts, among which some are well-known *FSHR*-responsive genes in Sertoli cells, including genes involved in cell cycle and survival regulation, such as cyclin D1. Comparable results were obtained in hypogonadal (*hpg*) mice, a convenient model to analyze the effect of exogenously administrated FSH on gene regulation in vivo [74,75]. Recently, the FSH-regulated gene expression landscape has been described not only in Sertoli cells from neonate or prepubertal rats but also in adults, when these cells fully provide metabolic and physical support of the germline [76]. As expected, in immature Sertoli cells, genes involved in cell growth and proliferation, metabolism, and the MAPK and Wingless-related integration site (Wnt) pathways were upregulated, whereas in mature cells, genes relevant to the differentiated function of Sertoli cells, involved in phagocytosis, cytoskeleton remodeling, glucose metabolism, and insulin signaling, were expressed. In this study, FSH was delivered with testosterone in a pulsatile fashion, expected to amplify the gene responses, as previously described [60]. This observation is very interesting because it suggests that hormone target cells decode a pulsatile signal vs. a monotonous signal in a quantitatively different manner. These transcriptomic analyses have provided an atlas of genes regulated by FSH signaling in vitro/ex vivo, but they indicate neither the transcription factors that could recognize the gene regulatory regions nor the upstream signaling pathways that could be involved. Systemic analyses of the global FSH-induced network from the receptor to the target genes will upgrade our understanding at the molecular level of FSH-induced biological responses. For example, the requirement of CREB in transcriptional regulation has to be reconsidered because not so many *FSHR*-responsive genes include a cAMP-responsive element (CRE) in their promoter regions. Hence, several PKA-mediated transcriptional responses could be mediated by other transcription factors, such as retinoic acid receptor α [77], a well-known regulator of germ cell development in the testis.

The anabolic role of FSH in Sertoli cells is well established; however, the whole FSH-induced proteome is not available yet. However, several reports have suggested that FSH regulates the translation efficacy of pre-existing mRNA in Sertoli cells from pre-pubertal rats. For example, FSH stimulates the mTOR/p70S6K pathway, inducing the phosphorylation of regulators of translation initiation, such as the scaffold proteins eukaryotic initiation factor 4B and G (eIF4B and eIF4G, respectively). These molecular rearrangements are linked to recruitment of specific mRNAs to the polysomes, such as the c-fos and vascular endothelial growth factor (VEGF) encoding mRNA [61]. In agreement with this trophic role, in transfected HEK293 cells expressing *FSHR*, the hormone enhances the translation of a particular subclass of mRNA, mainly encoding proteins of the translational machinery, the 5′TOP encoding mRNAs. However, polysome-bound mRNA identified by the riboTag technology in adult testis in vivo detected negligible changes in the amount of FSH-induced mRNAs [78].

MiRNAs constitute a bona fide network, intertwined with cell signaling networks in the cell. In this line, Sertoli cell-selective knock-out in mice of the gene encoding Dicer, an enzyme involved in miRNA processing, has unraveled the role of miRNAs in regulating the expression of genes essential for meiosis and spermiogenesis [79]. A rat model, where FSH and testosterone action was suppressed in vivo, has been created to identify the miRNA network at spermiation [80], a stage that is particularly sensitive to hormone regulation. Four of the regulated miRNAs that came out from a miRNA microarray analysis were complementary to the PTEN mRNA, and the hormonal input would lead to the degradation or synthesis inhibition of these miRNAs, then stabilizing PTEN at spermiogenesis [80]. Interestingly, the PTEN protein level is massively enhanced following FSH cell stimulation in vitro, leading Sertoli cells to achieve terminal differentiation [71]. Since FSH enhances PTEN protein level within minutes, the mechanisms involved probably occur post-transcriptionally, an assumption consistent with the hormone-induced degradation of miRNA that prevents the accumulation of PTEN locally in Sertoli cells. Hence, miRNA networks might regulate the compartmentalization of FSH signaling components in Sertoli cells and control the kinetics of these intracellular reactions.

Sertoli cells represent a paradigmatic model of transition between a proliferative state and commitment to differentiation that is controlled by FSH [81,82] and by other factors like thyroid hormones [83], which both act antagonistically. Seminal knock-out experiments in mice have demonstrated that thyroid hormones are the master signal that arrests Sertoli growth via the α isoform of the thyroid hormone receptor [84,85,86]. Local autocrine/paracrine secretion of other factors, such as glial cell-derived neurotrophic factor (GDNF) [63], *activin A*, and insulin-like growth factor (IGF)-I [87] also synergize, or at least complement, FSH action. In vivo ablation of the IGF system (both IGF-R and Ins-R knock-out) have convincingly entrenched that it is required for FSH-mediated mitogenic action in the pre-pubertal mouse [88]. Based on previous observations, it can be assumed that FSH, IGF-I, and triiodothyronine (T3) directly impact on cell cycle regulators. For instance, both FSH and IGF-I induce the expression of cyclin D1 and D2 [59,88], promoting the cell cycle progression through the G1 phase, and downregulating the transcription of two cell cycle inhibitors, namely, the p15^Ink4^ gene [88] and, presumably, the p21^Cip^ gene [89], via p53 dephosphorylation [90]. This action on the positive regulators of the cell cycle is counteracted by T3, which downregulates the expression of cyclin-dependent kinase 4 [91], whose expression is upregulated by FSH in granulosa cells [92]. T3 likely inhibits molecular events occurring in early G1, and not later on, because neither the E, A and cyclins B nor cyclin-dependent kinase 2 (Cdk2) are modulated by T3 (Figure 3) [84]. T3 acts not only through thyroid hormone receptor (TR)α1 but also through mitochondrial p43 receptors [84,85,91].

As indicated above, a number of studies established that FSH and testosterone are both mandatory for spermatogenesis to proceed appropriately. Interestingly, recent investigations on the respective role of FSH and testosterone by cell-specific knock-out of their respective receptor have shown that the extent of the Sertoli cell population is primarily determined by FSH, enabling the progression of germ cells through meiosis in concert with testosterone [33,93]. The steroid is also involved in suppression of Sertoli cell proliferation by targeting inhibitors of cell cycle progression [94].

## 4. Genetic Regulation of FSH Action on Spermatogenesis

The gene encoding the *FSHR* is located on chromosome 2p21, including ten exons and nine introns [95]. The first nine exons encode the extracellular receptor domain, mainly involved in ligand binding. Exon 10 encodes the hinge region, the seven transmembrane-spanning domains and the intracellular, carboxy-terminal tail [3]. Once FSH binds the extracellular FSHR domain, the receptor undergoes conformational changes, activating several intracellular signaling pathways [96], as described above. Thus, spatial conformations of FSHR specifically linked to single nucleotide polymorphisms (SNPs) may modulate the receptor functionality. More than two thousand SNPs have been detected within coding and non-coding regions of the *FSHR* gene, and two of them have been repeatedly evaluated. The adenosine to guanine change at position 919 from the start codon (c.919A > G; rs6165) and the c.2039A > G (rs6166) SNPs are located in exon 10 of the *FSHR* gene, leading to the amino acid changes p.T307A and p.N680S, respectively, in the receptor protein chain [97].

The physiological impact of *FSHR* SNPs was first described in men [98], although it was extensively studied in women undergoing assisted reproduction. The *FSHR* p.N680S homozygous S (S/S) genotype was related to higher FSH basal serum levels and to a higher number of FSH ampoules required to achieve similar estradiol levels during ovarian stimulation compared to the p.N680S homozygous N (N/N) genotype in 161 ovulatory women [99]. This was the first demonstration of a possible role of SNPs on female reproduction, suggesting that the presence of the S/S variant renders the FSHR less sensitive to FSH [97]. In vitro studies on human primary granulosa cells highlighted that the polymorphic variant of the receptor mediates different kinetics of the response to FSH stimulation [45]. Indeed, the homozygous *FSHR* S variant leads to a slower increase of intracellular cAMP than the p.N680S homozygous N receptor variant, although similar *plateau* levels were achieved after one hour of treatment [100]. Moreover, the *FSHR* genotype-specific cAMP kinetics impacts the activation of other molecules, such as ERK 1/2 and CREB, the expression of target genes, and progesterone production [100], reflecting clinical data demonstrating that the *FSHR* SNP p.N680S is a marker of ovarian response to FSH [101]. Although the role of this SNP on female reproduction was further confirmed in other studies [102], several years passed before its potential role in man was demonstrated. In 2012, the first demonstration of a modulatory activity of the *FSHR* p.N680S SNP on male fertility was produced, reporting a lower testicular volume in two large cohorts of the p.N680S homozygous S carriers compared to the N/N and heterozygous (N/S) patients [103,104]. Since a relatively high sample size was required for consistently detecting the physiological effect in men, it may be concluded that the p.N680S SNP has a weak impact on male fertility [105].

The *FSHR* promoter SNP, falling 29 nucleotides upstream of the transcriptional start codon, −29G > A (rs1394205), is another common polymorphism potentially impacting FSHR function [106,107] through the modulation of the gene transcriptional activity [108,109]. Interestingly, other SNPs involved in the gonadal response to FSH were identified within the FSHβ-encoding gene promoter. The *FSHB* −211G > T (rs10835638) is linked to reduced gene transcription and serum FSH levels [109]. Many authors have evaluated the *FSHB* c.−211G > T effect in both sexes, finding that, in men, it is associated with lower testicular volume, sperm count, testosterone, and LH serum levels [104]. The gonadal response to FSH would depend on FSHR functioning and number in the cell membrane, together with the amount of ligand available. Therefore, the combination of SNPs modulating the *FSHR* and *FSHB* gene transcription and receptor functioning may be predictive of the gonadal response to the hormone in vivo. A recent case–control study evaluated three *FSHR* SNPs (c.919A > G, c.2039A > G, and c.−29G > A) in 255 infertile men and 340 healthy controls [110]. Although the frequency of allelic variants was similar between the two groups, a specific haplotype was detected more frequently in fertile men (i.e., −29G > A G allele, c.919A > G A allele, and c.2039A > G A allele) [110]. Moreover, a meta-analysis of twelve studies available in the literature confirmed that the presence of either c.919A > G G (p.T307A A) or c.2039A > G G (p.N680S S) alleles is associated to an increased risk of male infertility [110]. This result suggests that different genetic models should be considered for inferring the combinatorial contribution of *FSHR* SNPs in male infertility, including dominant, co-dominant, and recessive models.

Previous studies examining potential associations between *FSHR* and *FSHB* SNPs and male fertility parameters have produced contradictory results, although they suggest that the FSH action on spermatogenesis is likely regulated by a combined effect of many SNPs [110]. Further evidence could be obtained by new studies in which complex haplotypes calculated using several SNPs falling even within genes not involved in the control of the FSH action. Moreover, available studies did not collect all fertility-related clinical parameters [110], limiting our current understanding of the impact of SNP combinations on male reproduction. The association between spermatogenetic phenotypes and genetic background could be necessary for developing patient-targeted therapeutic approaches.

## 5. Therapeutic Options to Improve Sperm Production

HH represents the main context of FSH application in males in clinical practice. This disease is due to congenital or acquired impairment of hypothalamic GnRH production and/or pituitary gonadotropin secretion [111,112,113,114]. The reduced gonadotropin stimulation results in low sex hormone secretion and sperm production. There is compelling evidence that HH patients benefit from either GnRH or gonadotropin replacement therapy, which results in an improvement of both spermatogenesis and androgenisation [115,116].

GnRH can be administered in a pulsatile fashion, subcutaneously via a portable infusion pump. This represents the most physiological approach to replacing gonadotropin stimulation of the testis. Pulsatile GnRH administration can restore the development of adult secondary sex characteristics, normal testosterone serum levels, and spermatogenesis [117,118,119,120]. However, in spite of being the most physiological option, this treatment is not often used outside highly specialized centers, mainly in the United States, due to the prolonged use of an external pump that can be inconvenient and costly to maintained.

Instead of GnRH administration, gonadotropins could be directly used in HH. In particular, clinical benefits have been reported using hCG alone or associated with FSH [121]. In this setting, a recent meta-analysis evaluated the effects of gonadotropin administration on sperm concentration in HH subjects [1]. Ten studies provided hCG alone, fourteen provided concomitant hCG and FSH treatment, and twenty-three introduced FSH at a variable time point after hCG monotherapy [1]. The comprehensive result showed a beneficial effect of all treatment regimens in almost 75% of patients, considering the appearance of at least one sperm at conventional semen analysis [1]. This effect was detected by evaluating a total of 897 HH patients and obtained with either hCG alone or hCG plus FSH, although a slightly significantly better result came from the combined action of the two gonadotropins [1]. This result confirms that the intra-testicular testosterone action is required for spermatogenesis, and FSH improves this clinical outcome synergistically. However, the relatively low HH prevalence represents the main obstacle in organizing large trials, which would allow us to establish the best gonadotropin replacement therapy for recovering spermatogenesis in this context. Moreover, HH treatment, either with GnRH or with gonadotropins, requires the use of injections that burden the patient, possibly limiting compliance and efficacy.

FSH is sometimes empirically proposed in men presenting idiopathic infertility, where the pathogenesis of infertility is not clearly identified. In such cases, a specific treatment is not applicable, and the exogenous stimulation of sperm production may be only empirical. Historically, first attempts to stimulate spermatogenesis were made by the administration of estrogen receptor antagonists (i.e., tamoxifen and clomiphene citrate). Estrogen receptor antagonists are used off-label in male infertility in some countries, especially in the United States. The pituitary release of endogenous FSH and LH may be increased by inhibiting the negative feedback mechanism using estrogen receptor modulators (SERMs). A recent meta-analysis of eleven randomized trials, enrolling men with idiopathic infertility, showed a higher pregnancy rate in couples in which the male partner was treated with SERMs with antagonistic activity at the estrogen receptor compared to couples with untreated men [122]. This result is reflected by a significant rise in sperm concentration and motility after SERM administration, confirming that sperm production may be improved by increasing gonadotropin stimulation [122]. However, this result has been obtained in a limited number of patients with heterogeneous clinical conditions, limiting the conclusion in favor of or against the application of this treatment. Accordingly, the Cochrane meta-analysis concluded that there is not enough evidence in favor of SERM application in male infertility [123]. The potential benefit of estrogen receptor antagonists on sperm parameters could be explained by endogenous FSH “overstimulation” and LH-mediated increase in intra-testicular testosterone concentration, increasing the synergic stimulation of FSH and LH on sperm production.

Exogenous FSH administration was suggested as a possible therapeutic option in infertile men with FSH serum levels within the normal ranges (i.e., below 8 IU/L), despite possible on-label only in few countries [124,125]. The rationale of this approach relies on the attempt to overstimulate spermatogenesis, an approach that was investigated in non-human primates. Indeed, FSH administration to monkeys significantly increased the number of type A pale [126] and B spermatogonia [127]. These studies indicated that spermatogenesis could be enhanced above the physiological rate and can be further stimulated by FSH administration. However, eventual benefits from supra-physiological FSH levels in humans were never clearly demonstrated. Interestingly, hCG did not have any effect on monkeys [126,127], suggesting the existence of species-specific regulation of the gonadal response among primates. Seventeen clinical trials (twelve prospective randomized clinical trials and five non-randomized studies) and four meta-analyses have been published so far, aiming at determining FSH efficacy in male idiopathic infertility. Interestingly, heterogeneous type and dosage of gonadotropin preparations were used (Table 1). Moreover, fourteen out of seventeen trials considered pregnancy rate as a primary endpoint, which is not exempt from many potential biases (i.e., the fertility status of the female partner). The efficacy of FSH in supporting spermatogenesis should be evaluated considering sperm parameters as the primary endpoint. Finally, most of these trials exhibit many biases, limiting the evidence in favor of the efficacy of FSH administration in male idiopathic infertility. The major limitations of these studies are the primary outcome chosen, heterogeneity of FSH dosages and duration (Table 1), as well as the inclusion criteria.

Meta-analyses provide an overview of studies published so far, overcoming the limited sample size in each trial. However, even meta-analyses are sometimes based on very limited numbers of patients. The Cochrane collaboration combined seven studies, suggesting a final spontaneous pregnancy rate increase in couples in which the male partner was treated with FSH [144,145]. However, these analyses did not highlight a significant pregnancy rate increase after ART [144,145]. In 2015, a second meta-analysis was performed, enlarging inclusion criteria and combining fifteen trials (both randomized and non-randomized) in which FSH or hMG was administered to idiopathic infertile male partners of couples attending ART [146]. This study suggested the benefit of this treatment on pregnancy rate, even outside the context of ART, due to a rise in total sperm numbers [146]. However, these works, combined together, have only a maximum of 482 patients treated by FSH/hMG, limiting the strength of these conclusions. Recently, the clinical efficacy of FSH administration in male idiopathic infertility has been reported, highlighting that gonadotropin dosage and sperm number increase are positively correlated [147]. Only one study with a pharmacogenetic design aimed at a priori identification of those infertile men expected to respond to FSH administration [105]. In this trial, FSH treatment improved sperm quality established by considering the sperm DNA fragmentation index in *FSHR* p.N680S homozygous N carrier men [105]. However, the FSH beneficial effect was not detected after a three-month therapy but rather after three more months of treatment interruption, suggesting that the current three-month-scheme is probably too short to obtain a sustained effect [105]. Similarly, a post-hoc analysis of a different clinical trial confirmed that pharmacogenomic of *FSHB* could be useful to predict FSH efficacy in idiopathic male infertility [130,148]. In summary, the few properly designed clinical trials have contrasting outcomes and results obtained by meta-analyses do not overcome the heterogeneity of these studies, preventing conclusions useful for clinical practice. A recent Italian nation-wide survey evaluated the idiopathic male infertility management, detecting an FSH prescription rate of 55% [5]. Although this study was not designed to detect FSH efficacy in male infertility, it showed a slight increase in semen parameters in about half of the treated men, confirming the need for properly designed larger clinical trials in this setting [5,52].

A further, interesting field of investigation is the role of FSH/gonadotropin treatment in cases of non-obstructive azoospermia undergoing sperm retrieval by testicular sperm extraction (TESE) and micro-TESE. The vast majority of observational trials and reviews suggest that elevated FSH serum levels should be considered negative predictive markers of sperm retrieval after TESE/micro-TESE. Given the (presumed) primary testicular inability to produce sperms in such cases, there is no clear rationale in proposing a further FSH stimulation in the presence of serum FSH levels above the normal range. However, some studies reported an increased rate of sperms extracted after TESE/micro-TESE in men with non-obstructive azoospermia treated with clomiphene citrate [149], hCG [150], hMG [149], and FSH [151,152]. These findings would suggest that the minimal, residual spermatogenic activity could be stimulated even in case of elevated serum FSH levels, at least in some patients. However, the role of FSH administration in men with non-obstructive azoospermia undergoing surgical procedures is not supported by sufficient evidence and remains experimental.

## 6. Future Perspectives

Future perspectives in the clinical use of gonadotropins in the male should consider the challenges still present in treating both HH and idiopathic infertility. Considering gonadotropin administration in HH men, several questions remain unsolved. First, many studies have demonstrated that hCG alone is sufficient to restore spermatogenesis and that FSH addition could only raise the sperm number [1,153]. Thus, intra-testicular testosterone seems able to stimulate spermatogenesis alone, overwhelming the FSH action on Sertoli cells. Moreover, no clear evidence is available on the right timing in which FSH should be added to hCG stimulation or which FSH dosage is required. Thus, specific trials aiming at dissecting the best therapeutic approach to HH men are still outstanding, focusing on the need, the dose and the timing of FSH addition to hCG. Of note, all available results on this topic used hCG to stimulate intra-testicular testosterone production rather than the physiological hormone LH. Although LH and hCG act on the same receptor, recent studies demonstrated different signal transduction pathways at the cellular level and different clinical outcomes in vivo, at least in women undergoing assisted reproduction [2,154]. The reason for using hCG instead of LH is more related to historical and practical reasons (availability, half-life, and costs) than to scientific evidence. Indeed, the first gonadotropin isolation dates back to the 1930s to 1950s and the preparation obtained, called human menopausal gonadotropin (hMG), was a 50/50 mixture of FSH and LH. Methodological improvements to obtain pure hormones have resulted in the effective production of highly purified FSH from the urine of postmenopausal women but LH is lost in the chromatographic procedure. Therefore, there is the need to add hCG (from the urine of pregnant women) to modern hMG preparations. Considering hCG potency and half-life, no attempts were made to obtain pituitary LH from urine in the certainty that the two gonadotropins had the same biological effect [2]. Now that recombinant gonadotropins are available, the new evidence on the differences between LH and hCG action should be considered and could open further perspectives in the potential therapeutic role of LH in stimulating testicular function.

Gonadotropin administration to HH men leads to a sperm concentration increase up to a mean value of 5.92 million per mL until pregnancy is obtained, after which this regimen is stopped [1]. Although this increase is not sufficient to reach the normozoospermic range according to the World Health Organization (WHO) (i.e., >15 million per mL) [155], it is largely compatible with fertility in these men [156,157,158,159,160,161]. Thus, the reason why pharmacological gonadotropin stimulation does not completely restore spermatogenesis in such patients, even after many months, sometimes years (insufficient duration? non-optimal stimulation protocols? hCG vs. LH?), is unclear, intriguing, and remains to be fully discovered. Similarly, future perspectives in this area should consider which is the best schedule of gonadotropin stimulation. Indeed, both FSH and LH are physiologically secreted by the pituitary gland in a pulsatile fashion, and this secretory rhythm is considered essential for fertility in all mammals, including in human [162]. Whether the current, non-pulsatile regimen of administration of exogenous gonadotropins is the most effective to stimulate the human testis in clinical practice remains unclear.

Another important issue to be developed is how to treat HH boys before puberty, as they are currently diagnosed very early. Indeed, the standard management of male HH adolescents remains testosterone administration in order to induce virilization and psycho-sexual maturation [163]. However, testosterone replacement therapy does not induce spermatogenesis, which, on the contrary, requires the administration of gonadotropins [164]. It is currently unclear whether puberty induction should be performed first with testosterone, followed by gonadotropins, or vice versa [165]. Recently, Zacharin et al. described complete pubertal development in 19 HH boys, treated with both hCG and FSH, with the acquisition of fertility and a marked improvement of quality of life, regardless of previous testosterone replacement [166]. Thus, it seems that gonadotropin administration is required to stimulate HH testicular development and this could be performed both before and after testosterone administration. However, what the right timing is to start gonadotropins’ stimulation to induce puberty remains controversial and requires properly designed trials.

In the setting of male idiopathic infertility, further pharmacogenomics studies are needed to prospectively evaluate how to select patients to be treated and to tailor FSH administration. To this purpose, pharmacogenomics evaluation of the *FSHR* and *FSHB* SNP role on male fertility could be useful. Indeed, since genetic haplotypes are associated with a reduced physiological stimulation of FSH on spermatogenesis, these patients could benefit from higher FSH dosages or longer treatment duration. On the other hand, most favorable genetic haplotypes could be sufficiently stimulated with the current empirical FSH therapeutic regimens. Thus, new, comprehensive approaches, connecting the genetic background to spermatogenesis and treatment response, are required [52].

Finally, new perspectives deal with the application of new compounds with gonadotropin activity. For example, corifollitropin alfa is a recombinant hormone obtained by combining the gonadotropin α subunit with a chimeric FSHβ subunit fused to the carboxy-terminal peptide (CTP) of the hCGβ-subunit. This compound has a longer half-life than FSH due to additional glycosylation sites in the CTP portion [167]. Since, in women undergoing ART, corifollitropin alfa produces a similar therapeutic response as recombinant FSH formulations [168], it is claimed that one injection of corifollitropin alfa could replace seven FSH daily administrations. In the male context, since gonadotropin regimens commonly impose long-term treatment with frequent injections, the use of corifollitropin alfa may result in fewer medication errors and improved compliance [169]. Replacement therapy with corifollitropin alfa has been evaluated in HH men, demonstrating similar effectiveness and safety to FSH [169]. On the other hand, this new compound has never been tested in the context of male idiopathic infertility. Thus, future research on the potential application of corifollitropin alfa in idiopathic male infertility should be conducted.

Further attempts have been made to develop long-lasting gonadotropins. Efforts made along these lines included the attachment of polyethylene glycol (PEG) to increase half-life, bioavailability, and biological activity. Covalent addition of PEG results in increased solubility and increased size, hence, reduced renal clearance and protection from proteolytic degradation. FSH conjugated to PEG retained FSH activity in vitro on bovine cumulus cells [170]. Interestingly, PEGylated FSH displayed improved bioavailability compared to FSH in rats [171]. Another strategy for prolonging hormone half-life consists in the development of sustained-release formulations using aluminum hydroxide gel suspension. This approach led to prolonged bioactivities in rabbits and cattle [172,173].

Single-chain gonadotropins with improved pharmacokinetics have also been reported (see [174] for a recent review). A pioneer study [175] demonstrated that a single-chain recombinant analog, in which the hCGβ-subunit was fused to the N-terminus of the gonadotropin α-subunit, displayed improved in vivo biopotency compared to heterodimeric hCG. This initial success led to the development of single-chain analogs of LH and FSH. They were engineered with the β-subunits oriented at the N-terminus of the α subunit and used the hCGβ CTP sequence as a linker, resulting in increased half-life in vivo. Interestingly, they had similar or even higher secretion in vitro and biological activity in vivo than their native hormone counterpart [176,177,178]. For instance, it was demonstrated that a single injection of single-chain FSH analog could stimulate estrogen production for 5 to 7 days in Rhesus monkeys [179]. As expected, the O-linked glycosylation sites of the carboxyl-terminal peptide (CTP) reduced hepatic clearance rates, contributing to increased serum half-life and increased biopotency of the single-chain analogs. Some analogs presenting dual FSH and LH activities were later developed. Such dual-active analogs could be relevant for the treatment of HH and infertile men. These constructs included FSHβ-CTP-CGβ-α and FSHβ-CTP-LHβ-CTP-α, and, when injected in sheep, elicited increased serum estradiol concentration, ovarian weight, and formation of corpora lutea [177,180,181]. The clearance rate of the dual-active FSHβ-CTP-LHβ-CTP-α was also significantly improved compared with endogenous ovine gonadotropin (1 to 2 h). Despite these promising indications, to date, no single-chain or dual-active gonadotropin analog has reached the clinical phase of development.

More recently, a fusion protein consisting of FSHα and β-subunits fused to immunoglobulin Fc fragments was constructed in order to benefit from the advantageous pharmacokinetic properties of immunoglobulins [182,183]. Female rats injected this FSH-Fc analog displayed significantly increased ovarian weight compared to FSH-treated animals.

In a radically different approach, low molecular weight (LMW) chemicals exerting agonistic actions at the FSHR also represent an intriguing alternative for the treatment of men in HH and infertility. Although such compounds do not present relatively long half-life in vivo, they could be developed for oral administration. Another advantage is the relatively limited cost of this type of molecule compared to biologicals such as recombinant gonadotropins. Hence, long term treatments with a high dosage could be envisioned without compromising the patient’s compliance or health system financial sustainability. Although developing LMW agonists to gonadotropin receptors has been very challenging compared to some other GPCRs, this is likely due to the peculiar nature of FSH interaction with the extracellular domain of FSHR and the subsequent chain of molecular events that lead to signal transduction. Various LMW modulators acting at FSHR have been reported over the years. Amongst them, the thiazolidinone (TDZ) series was identified by combinatorial screening [184]. Some TDZ derivatives exhibit agonistic/positive allosteric modulation (PAM) at the FSHR [185]. Indeed, some TDZs are able to stimulate FSH-induced signaling in the absence of FSH and are able to induce folliculogenesis in immature rats. Despite such promising results, these compounds never reached the market due to unfavorable pharmacokinetic parameters [186]. Other compounds from the TDZ series showed the ability to switch FSHR coupling from Gα_s_ to Gα_i_ without the functional consequences for reproduction being clarified to date [187]. Another class of LMW ligands capable of binding at the FSHR is the benzamide series. These compounds show selectivity to FSHR compared to TSHR and LHCGR. In the presence of a physiological concentration of FSH (EC_20_), some benzamide derivatives enhanced the FSH-mediated activity on the receptor, behaving as PAM [188]. Similarly, the dihydropyridine compound Org 24444-0 was also reported to increase FSH-induced cAMP production in vitro through a PAM mechanism of action. Interestingly, Org 24444-0 reproduces the effects of FSH on follicle maturation in vivo [189]. Other LMW ligands have also been identified for their ability to turn off FSHR signaling. Among them, tetrahydroquinolines showed potent inhibitory effects on cAMP production without impairing FSH binding. Unfortunately, no effect was observed in vivo [190]. More recently, the ADX series included negative allosteric modulators (NAM) acting at the FSHR. In particular, ADX61623 shows decreased FSHR-induced cAMP production and improved FSH binding affinity. Furthermore, when applied to primary granulosa cells, ADX61623 decreased progesterone but not estradiol production [191]. Two other compounds from the ADX series were later reported as biased NAMs at the FSHR: ADX68692 and ADX68693. ADX68692 decreases cAMP, progesterone, and estradiol production in vitro in rat granulosa cells. However, ADX68693 shows similar inhibitory action on progesterone while it does not impair estradiol production in the same model system. Interestingly, in vivo, ADX68692 efficiently reduces the number of oocytes recovered in mature female rats whereas ADX68693 does not [192]. It is important to note, however, that these two compounds also affect LHCGR signaling, highlighting promiscuity among the gonadotropin receptors for this ADX series [193]. Despite these promising developments, no LMW compound has reached clinical phases so far. Further development will be needed to develop an orally active LMW FSHR agonist.

Whereas none of these different strategies to improve bioactivity/bioavailability/administration has been tested clinically yet, they do open promising research avenues for the development of gonadotropin therapeutics that are better adapted to the treatment of men in HH and infertility.

## 7. Conclusions

In conclusion, here we highlight how FSH acts in the sophisticated and not completely understood regulatory process resulting in male fertility. Human spermatogenesis is regulated by FSH and LH-dependent intra-testicular testosterone, with synergistic, partially overlapping mechanisms. Starting from these notions, FSH administration is proposed as a potentially effective therapeutic approach to male infertility. The exogenous FSH administration seems to improve sperm production, although it remains unclear whether it acts as replacement therapy or overstimulating treatment. This latter concept is supported by experimental demonstrations that spermatogenesis could be boosted beyond its physiological rate. However, the FSH-dependent overstimulation on spermatogenesis must still be verified in infertile men with properly designed clinical trials. Similarly, new scientific evidence is needed to confirm the efficacy of FSH administration in male infertility. A really powerful, phase 2 clinical trial is urgently required to produce evidence about FSH therapeutic potential. In this setting, the pharmacogenomic basis of FSH response, as well as the most effective dose and duration, must be addressed.

## Figures and Tables

**Figure 1 jcm-09-01014-f001:**
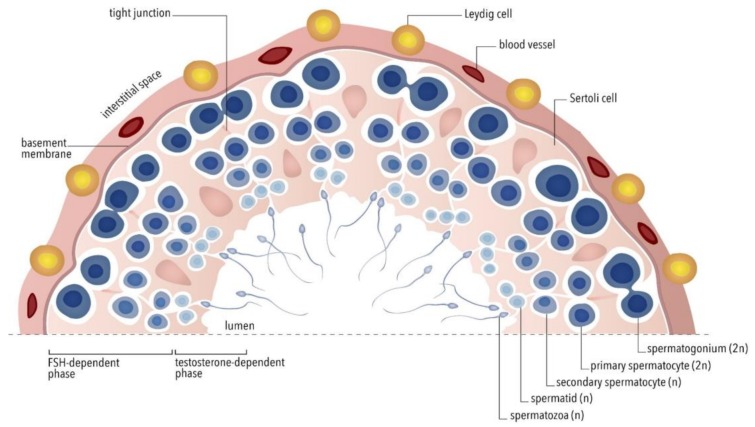
Schematic representation of a hemi-section of a seminiferous tubule. From the basal membrane to the lumen, spermatogenesis stages are reported up to the mature spermatozoa. This complex process is regulated synergistically by follicle-stimulating hormone (FSH) acting on Sertoli cells and by intra-testicular testosterone, produced by Leydig cells under luteinizing hormone (LH) stimulus. (FSH = follicle-stimulating hormone; *n* = haploid cell; *2n* = diploid cell)

**Figure 2 jcm-09-01014-f002:**
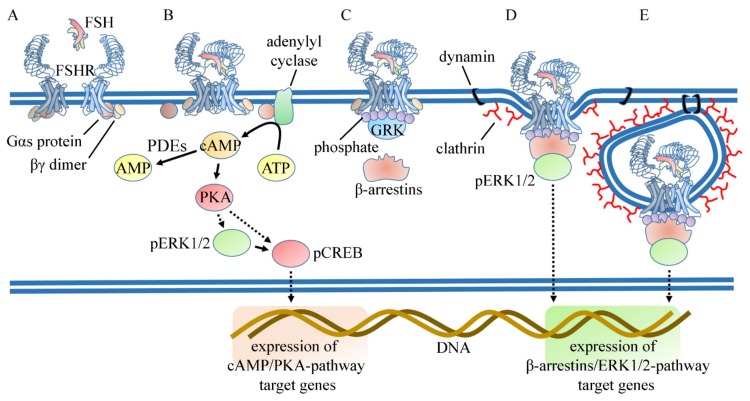
Early FSH-dependent signaling and receptor internalization. (**A**) FSHR extracellular domain is exposed to FSH binding in the cell membrane. (**B**) FSHRs may form multimers in the cell surface and, upon FSH binding, undergo conformational changes inducing activation of the Gα_s_ protein and subsequent ATP to cAMP conversion by the adenylyl cyclase enzyme. The downstream PKA, pERK1/2, and pCREB activation target CRE sequences, triggering gene expression. (**C**) The GRK enzyme is recruited via a mechanism involving the βγ dimer of the G protein and mediates FSHR phosphorylation at specific intracellular sites. (**D**) β-arrestins form a signaling module associated with FSHR and ERK1/2, mediating the assembly of clathrin-coated pits and internalization of the receptor. Besides this β-arrestin/clathrin-related mechanism, the GTPase dynamin may participate in the internalization of FSHR. (**E**) FSH complexed with the receptor is internalized and routed to endosomal compartments sustaining the prolonged activation of signaling cascades.

**Figure 3 jcm-09-01014-f003:**
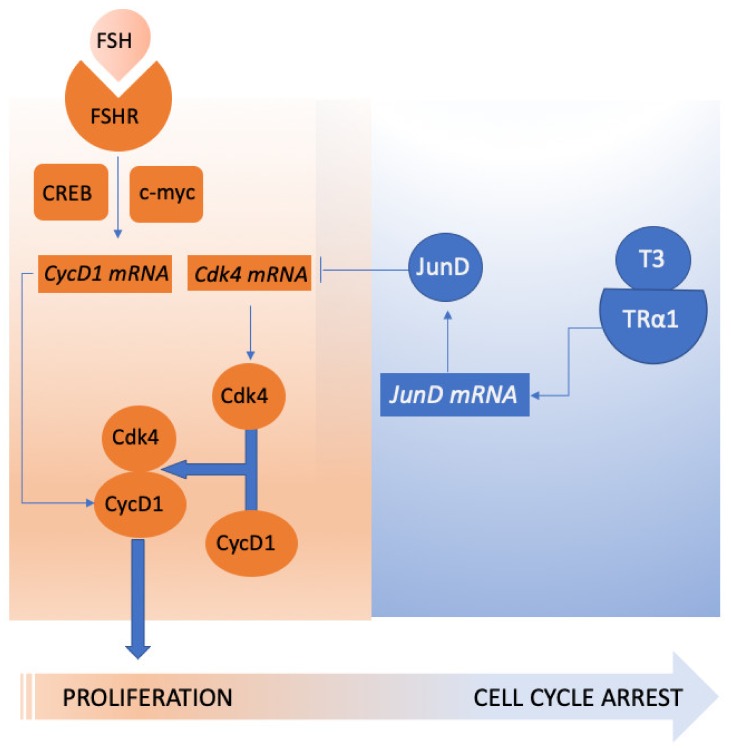
Triiodothyronine (T3) counteracts the mitotic response to FSH, by inhibiting the transcription of Cdk4, via the JunD transcriptional regulator. Thus, the association of Cdk4 to FSH-induced CycD1 is hampered, and Sertoli cell mitoses cease.

**Table 1 jcm-09-01014-t001:** Clinical trials available in the literature evaluating the efficacy of follicle-stimulating hormone (FSH) in male idiopathic infertility. Studies are classified considering the study design and the therapeutic dosage applied.

FSH Doses	Treatment Duration	FSH Total Amount	Authors	FSH Doses	Treatment Duration	FSH Total Amount	Authors
**RCT**							
Recombinant FSH				Urinary-derived FSH			
50 IU on alternate days	12 weeks	2250 IU	Foresta et al. (2002) [128]	50 IU on alternate days	12 weeks	2250 IU	Ding et al. (2015) [129]
100 IU on alternate days	12 weeks	4500 IU	Foresta et al. (2002), Foresta et al. (2005) [128,130]	100 IU on alternate days	12 weeks	4500 IU	Ding et al. (2015) [129]
		6750 IU	Colacurci et al. (2012) [131]				
150 IU on alternate days	12 weeks	6750 IU	Simoni et al. (2016), Foresta et al. (2009) [105,132]	75 IU daily (150 IU on alternate days)	Not reported	Not reported	Ben-Rafael et al. (2000), Matorras et al. (1997) [133,134]
		5400 IU	Selice et al. (2011) [135]		13 weeks	6825 IU	Knuth et al. (1987)^*§^ [136]
150 IU daily (300 IU on alternate days)	12 weeks	12600 IU	Kamischke et al. (1998)^§^ [137]	200 IU on alternate days	12 weeks	9000 IU	Ding et al. (2015) [129]
	16 weeks	18000 IU	Paradisi et al. (2006)^§^ [138]	150 IU daily (300 IU on alternate days)	12 weeks	12600 IU	Ding et al. (2015), Ben-Rafael et al. (2000) [129,133]
**Non randomized, retrospective trials**							
Recombinant FSH				Urinary-derived FSH			
150 IU on alternate days	12 weeks	5400 IU	Caroppo et al. (2003) [139]	75IU on alternate days	12 weeks	3375 IU	Foresta et al. (2000) [140]
				75 IU daily	12 weeks	6300 IU	Ashkenazi et al. (1999) [141]
					4 weeks	2100 IU	Bartoov et al. (1994) [142]
				150 IU daily	12 weeks	12600 IU	Baccetti et al. (2004) [143]

FSH: follicle-stimulating hormone; RCT: randomized controlled clinical trial. * This study applied human chorionic gonadotropin (hCG) plus human menopausal gonadotropin (hMG). § In these studies, the control group was treated with placebos in a double-blind study design.

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
