# Peer review of "Follicle-Stimulating Hormone (FSH) Action on Spermatogenesis: A Focus on Physiological and Therapeutic Roles"

_jcm, 2020, doi:10.3390/jcm9041014_

Round 1
Reviewer 1 Report
This manuscript provides a rather detailed review on the function of FSHspermatogenesis. It shows a nice combination of results obtained in rodents and primates. This review will be quite usefulfor people working in this field. As described in more detail below, there is an important omission in the data discussed, which is that data on successful stimulation of sperm production by increasing FSH levels in monkeys, obtained by 2 different groups, are not mentioned. As this concerns a major question discussed in this in this review, the papers describing these results should be fully taken into account.
lines 93-94: Refs 23 and 24 do not reliably support the described positive effects of FSH administration on germ cell numbers in mice. In ref 23 it concerns testosterone administration that induces higher levels of FSH in GnRH immunized mice and the joint effect of testosterone and FSH on germ cell numbers. Ref 24 apparently consists of a remark in the Discussion of the referenced paper. Therefore this sentence should be deleted.
Legend of fig. : It should be added that this is a schematic representation that is not a naturally occurring cell association. This is a gathering of germ cell types that will never occur in any mammalian species. Alternatively, the authors could redraw the figure and present a normal epithelial stage.
Lines 381-386: This is a complex sentence that is way too long. This notion should be presented in a couple of sentences.
Lines 452-498: This part is about the question whether or not FSH cab used to boost sperm production in men, even when FSH levels are in the normal range. Surprisingly, the positive results obtained by 2 independent groups (groups of Dirk de Rooij and of Tony Plant) are not mentioned at all. The results of these groups should be fully discussed and when the authors do not agree with the conclusions drawn from these experiments, they should explain why. See:
-FSH stimulates spermatogenesis in the adult monkey. Van Alphen et al. Endocrinology 1988. 123(3):1449-55. PMID 3136008
A selective monotrpic elevation of FSH, but not of LH, amplifies the proliferation and differentiation of spermatogonia in the adult rhesus monkey (Macaca mulatta). Simorangkir et al. Hum.Reprod. 2009. 24(7): 1584-95. PMID 19279035
Author Response
Reviewer 1
This manuscript provides a rather detailed review on the function of FSH spermatogenesis. It shows a nice combination of results obtained in rodents and primates. This review will be quite useful for people working in this field. As described in more detail below, there is an important omission in the data discussed, which is that data on successful stimulation of sperm production by increasing FSH levels in monkeys, obtained by 2 different groups, are not mentioned. As this concerns a major question discussed in this in this review, the papers describing these results should be fully taken into account.
ANSWER: Thank you for your suggestion. We considered the two manuscripts that you cited as reported below.
Comment 1
lines 93-94: Refs 23 and 24 do not reliably support the described positive effects of FSH administration on germ cell numbers in mice. In ref 23 it concerns testosterone administration that induces higher levels of FSH in GnRH immunized mice and the joint effect of testosterone and FSH on germ cell numbers. Ref 24 apparently consists of a remark in the Discussion of the referenced paper. Therefore this sentence should be deleted.
ANSWER: We deleted the sentence: “FSH-treated mice show an increased number of germ cells entered in the early stages of spermatogenesis up to round spermatids” as well as the two references.
Comment 2
Legend of fig. 1: It should be added that this is a schematic representation that is not a naturally occurring cell association. This is a gathering of germ cell types that will never occur in any mammalian species. Alternatively, the authors could redraw the figure and present a normal epithelial stage.
ANSWER: We have changed the figure caption accordingly, thanks to the Reviewer.
Comment 3
Lines 381-386: This is a complex sentence that is way too long. This notion should be presented in a couple of sentences.
ANSWER: We changed the sentence as follows: “The gonadal response to FSH would depend on FSHR functioning and number in the cell membrane, together with the amount of ligand available. Therefore, the combination of SNPs modulating the FSHR and FSHB gene transcription and receptor functioning may be predictive of the gonadal response to the hormone in vivo”.
Comment 4
Lines 452-498: This part is about the question whether or not FSH can used to boost sperm production in men, even when FSH levels are in the normal range. Surprisingly, the positive results obtained by 2 independent groups (groups of Dirk de Rooij and of Tony Plant) are not mentioned at all. The results of these groups should be fully discussed and when the authors do not agree with the conclusions drawn from these experiments, they should explain why. See:
-FSH stimulates spermatogenesis in the adult monkey. Van Alphen et al. Endocrinology 1988. 123(3):1449-55. PMID 3136008
A selective monotrpic elevation of FSH, but not of LH, amplifies the proliferation and differentiation of spermatogonia in the adult rhesus monkey (Macaca mulatta). Simorangkir et al. Hum.Reprod. 2009. 24(7): 1584-95. PMID 19279035
ANSWER: Thanks for reminding us of these two studies, which are indeed very important to underline the concept of FSH potential for male infertility. They are now commented as follows: “Exogenous FSH administration was suggested as a possible therapeutic option in infertile men with FSH serum levels within the normal ranges (i.e. below 8 IU/L), despite possible on-label only in few countries [125,126]. The rationale of this approach relies on the attempt to overstimulate spermatogenesis, an approach that was investigated in non-human primates. Indeed, FSH administration to monkeys significantly increased the number of type A pale [127] and B spermatogonia [128]. These studies indicated that spermatogenesis could be enhanced above the physiological rate can be further stimulated by FSH administration. However, eventual benefits from supra-physiological FSH levels in humans were never clearly demonstrated. Interestingly, hCG did not have any effect in monkeys [127,128], suggesting the existence of species-specific regulation of the gonadal response among primates”.

Reviewer 2 Report
The Authors are proposing an interesting and well written narrative review on FSH role in spermatogenesis and its potential therapeutic action in men. However, given the availability of many other reviews on this topic, to enhance the relevance of the present paper, I would suggest to add an additional paragraph on rationale and limits of using FSH to improve spermatogenesis in men with non-obstructive azoospermia.
Minor comments
Page 2, line 56 The statement “Although the full restoration of normal semen parameters cannot be achieved in HH men” is questionable. Response to FSH is variable in HH men, depending on causes and (prepubertal vs post-pubertal) onset of the disease; however, normal sperm parameters can be achieved in a subset of these patients [Andrology 2016; 4:87-94].
Table 1 reports data from studies that are not listed in the References section: the Authors are, therefore, requested to add the missing references (number 175 through 185 as provided by table 1).
Author Response
Reviewer 2
The Authors are proposing an interesting and well written narrative review on FSH role in spermatogenesis and its potential therapeutic action in men. However, given the availability of many other reviews on this topic, to enhance the relevance of the present paper, I would suggest to add an additional paragraph on rationale and limits of using FSH to improve spermatogenesis in men with non-obstructive azoospermia.
ANSWER: The rationale and the limits of using FSH to improve human spermatogenesis are presented at page 13, within the paragraph “Therapeutic options to improve sperm production”. We have now added some sentences to make it clearer.
Minor comments
Comment 1
Page 2, line 56 The statement “Although the full restoration of normal semen parameters cannot be achieved in HH men” is questionable. Response to FSH is variable in HH men, depending on causes and (prepubertal vs post-pubertal) onset of the disease; however, normal sperm parameters can be achieved in a subset of these patients [Andrology 2016; 4:87-94].
ANSWER: We changed the sentence as follows: Although the full restoration of normal semen parameters is generally not achieved in HH men (depending on HH causes) [4], gonadotropin treatment is sufficient to stimulate spermatogenesis [1] and provides a rationale for using, even if empirically, FSH with or without hCG, in male idiopathic infertility, considered as a form of “functional” hypogonadism
Comment 2
Table 1 reports data from studies that are not listed in the References section: the Authors are, therefore, requested to add the missing references (number 175 through 185 as provided by table 1).
ANSWER: we corrected the references within table 1
Round 2
Reviewer 1 Report
The authors have responded satisfactorily to my comments. I have no further comments.
Author Response
Thank you
Reviewer 2 Report
The Authors have revised the manuscript, however there is one point that need to be clarified. In my previous reviewer comment, I wrote "However, given the availability of many other reviews on this topic, to enhance the relevance of the present paper, I would suggest adding an additional paragraph on rationale and limits of using FSH to improve spermatogenesis in men with non-obstructive azoospermia". The Authors haven't added the requested paragraph, probably because the comment was not that clear. In more clear terms, my request is to discuss the available evidences about the effect of FSH on sperm retrieval rate in patients with non-obstructive azoospermia undergoing surgery (TESE/microTESE), given that the advantages of such a treatment are currently under debate.
A discussion on the effective role of FSH treatment to improve sperm retrieval rates in patients with NOA would really add to the manuscript, as most of the available reviews haven't deal with it.
Author Response
Comment 1
The Authors have revised the manuscript, however there is one point that need to be clarified. In my previous reviewer comment, I wrote "However, given the availability of many other reviews on this topic, to enhance the relevance of the present paper, I would suggest adding an additional paragraph on rationale and limits of using FSH to improve spermatogenesis in men with non-obstructive azoospermia". The Authors haven't added the requested paragraph, probably because the comment was not that clear. In more clear terms, my request is to discuss the available evidences about the effect of FSH on sperm retrieval rate in patients with non-obstructive azoospermia undergoing surgery (TESE/microTESE), given that the advantages of such a treatment are currently under debate. A discussion on the effective role of FSH treatment to improve sperm retrieval rates in patients with NOA would really add to the manuscript, as most of the available reviews haven't deal with it.
ANSWER: We added the following paragraph at the end of the “Therapeutic options to improve sperm production” section, following your suggestions: “A further, interesting field of investigation is the role of FSH/gonadotropin treatment in cases of non-obstructive azoospermia undergoing sperm retrieval by testicular sperm extraction (TESE) and micro-TESE. The vast majority of observational trials and reviews suggests that elevated FSH serum levels should be considered negative predictive markers of sperm retrieval after TESE/micro-TESE. Given the (presumed) primary testicular inability to produce sperms in such cases, there is no clear rationale in proposing a further FSH stimulation in the presence of serum FSH levels above the normal range. However, some studies reported an increased rate of sperms extracted after TESE/micro-TESE in men with non-obstructive azoospermia treated with clomiphene citrate [150], hCG [151], hMG [150] and FSH [152,153]. These findings would suggest that the minimal, residual spermatogenic activity could be stimulated even in case of elevated serum FSH levels, at least in some patients. However, the role of FSH administration in men with non-obstructive azoospermia undergoing surgical procedures is not supported by sufficient evidences and remains experimental.”